# Simulation Analysis of Signal Conditioning Circuits for Plants' Electrical Signals

**Mirella Carneiro** [1], **Victor Oliveira** [1], **Fernanda Oliveira** [1], **Marco Teixeira** [2,*] **and Milena Pinto** [3]

1   Electrical Engineering Program, COPPE, Federal University of Rio de Janeiro, Rio de Janeiro 21941-972, Brazil
2   Department of Software Engineering, Federal University of Technology-Parana (UTFPR),
    Curitiba 80230-901, Brazil
3   Department of Electronics Engineering, Federal Center for Technological Education of Rio de Janeiro,
    Rio de Janeiro 20271-110, Brazil
*   Correspondence: marcoteixeira@utfpr.edu.br

**Abstract:** Electrical signals are generated and transmitted through plants in response to stimuli caused by external environment factors, such as touching, luminosity, and leaf burning. By analyzing a specific plant's electrical responses, it is possible to interpret the impact of external aspects in the plasma membrane potential and, thus, determine the cause of the electrical signal. Moreover, these signals permit the whole plant structure to be informed almost instantaneously. This work presents a brief discussion of plants electrophysiology theory and low-cost signal conditioning circuits, which are necessary for the acquisition of plants' electrical signals. Two signal conditioning circuits, which must be chosen depending on the signal to be measured, are explained in detail and electrical simulation results, performed in OrCAD Capture Software are presented. Furthermore, Monte Carlo simulations were performed to evaluate the impact of components variations on the accuracy and efficiency of the signal conditioning circuits. Those simulations showed that, even after possible component variations, the filters' cut-off frequencies had at most 4% variation from the mean.

**Keywords:** plant electrophysiology; electrical signals; information acquisition; simulation software; electronic instrumentation





## 1. Introduction

Plants are organisms aware of diverse factors in the habitat they are placed. Furthermore, they continuously adapt their metabolism and growth in response to environmental changes. Due to this adaptation mechanism, plants have developed techniques to react right after they detect habitat modification aspects and external stimuli. They respond to these factors by transmitting electrical responses through their structure. Plants' electrical activities are related to transient modifications in the plasma membrane potential [1]. The flow of ions and the activation of ion channels induces a transient and local change in the potential of the cell membrane. Taking into account the main reason for this change is that all cells (mainly root cells associated with ions uptake) hold the whole time ions essentially crossing the plasma membrane [2,3].

Distinct sorts of disturbances, like an abrupt light variation, soil moisture content, and leaf burning, can generate these specific electrical signals in living plant cells, according to [1,4,5]. In contrast to chemical signals, electrical responses originated by these stimuli can conduct information quickly over long distances, from the top of the stem to the roots, in either direction [6]. Besides, once initiated, these responses spread to adjoining excitable cells. The coordination of internal processes and their balance with the environment is connected to plant cells' excitability [7].

Plants have four different types of electrical signals, which are: *(i)* action potentials (APs); *(ii)* variation potentials (VPs); *(iii)* local electrical potentials (LEPs); and *(iv)* system potentials (SPs).

LEP is a local electrical signal generated from natural changes related to the environment, such as luminosity, soil nutrients, and air humidity. These changes cause a sub-threshold electrical response in plants [8]. SP was detected in the plant leaves after caterpillar feedings. Besides, it is a long-distance signal with duration and amplitude dependent on the stimulus [9]. AP is induced by a non-damaging disturbance to the plant (electrical, mechanical stimulus, or thermal shock [6]) and is characterized by transmitting information over long distances along the plant in a short amount of time. VP is caused by harmful stimuli to the plant, such as burning and cutting. The plant type and the disturbance's intensity have influence on the VP signal shape and magnitude [10].

The way some characteristics of the electrical signal, like amplitude, duration and speed of the electrical signal behaves while propagating through the plant structure depends primarily on the type of the signal, i.e., if it is an AP, VP, LEP ou SP. Since each signal has got their own peculiarities, which will be explained further in Section 2.

Furthermore, two methods to measure electrical potential in plants can be employed: extracellular and intracellular [11].

According to [12], real-time monitoring of these electrical signals enable the user to be informed about what happens in the habitat where the plants are placed. With this information, it is possible to identify the presence of landslides [13], acid rain [14], an increase in air pollution, whether the plant receives too much light or if pests are attacking a certain plant in the plantation [9]. Figure 1 shows the necessary steps to acquire plants' electrical responses. This work addresses the fifth step: Signal Conditioning Circuit.

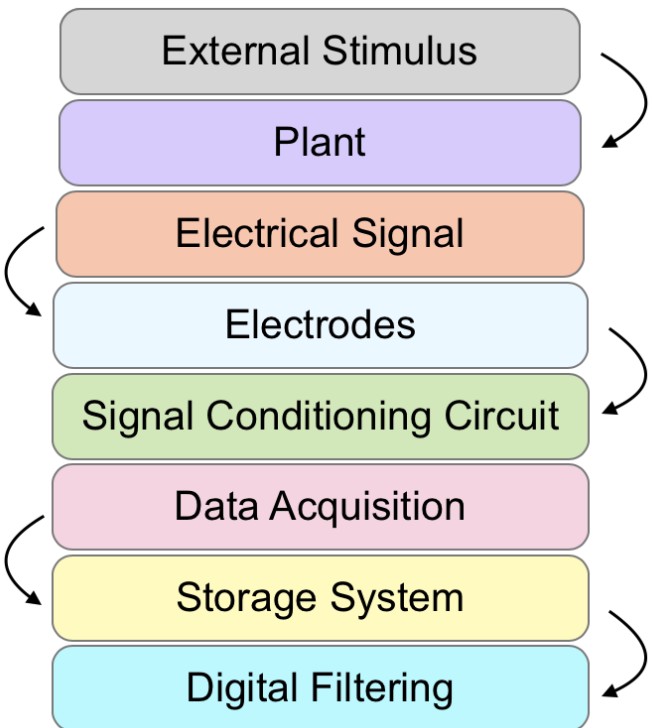

**Figure 1.** Flowchart of the steps used to acquire plants electrical signals. Adapted from [3,12].

Usually, when a sensor is used to measure a signal, the sensor reads not only the desired stimulus, but also noise. Furthermore, the measured electrical signal amplitude from the plant may not be large enough for proper data acquisition. Most signal conditioning circuits purpose is to filter, to reduce noise, and/or amplify the original signal in order to enable data acquisition.

When measuring plants' electrical responses, an analog-to-digital converter (ADC) is used to convert the waveform of the electrical signal into digital data. Furthermore, it is necessary to carefully choose an ADC with appropriate sampling frequency according to

the measured signal. In addition, the greater the input impedance of the ADC, the closer the ADC input signal value is to the signal conditioning circuit output. Therefore, the ADC input impedance has to be at least 100 times greater than the output impedance of the signal conditioning circuit. Besides, since environmental factors influence the plants' signals, it is important to employ an environmental parameters-acquiring system to measure such factors. Furthermore, the plant, along with the unshielded components of the measuring system, must be placed inside a Faraday cage, in order to improve the signal-to-noise ratio (SNR) of the measured signal [12]. The complete acquisition system is shown in Figure 2.

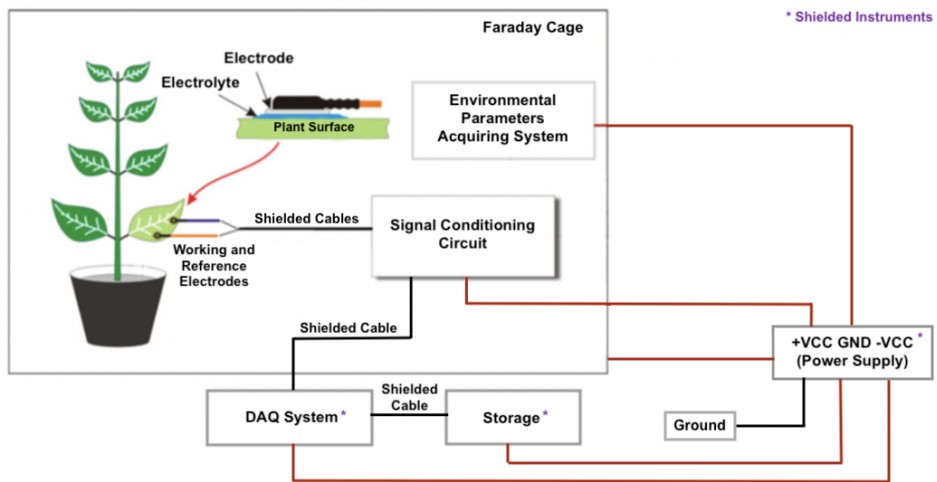

**Figure 2.** The complete acquisition system for measuring plants electrical responses [12].

More details about plants acquisition system can be found in [3,12].

### 1.1. Related Works and Main Contributions

In most of the works carried out in the area of measuring electrical signals in plants, the equipment applied in the process is expensive, as shown in [6,7,14–16]. Besides, there are only a few authors that design their own signal conditioning circuits. Most authors employ ready-for-use instruments to perform this task, which contributes to the increased cost of capturing electrical responses emitted by plants. A requirement of the equipment (or the developed signal conditioning circuit) that reads the plant signal measured by the sensor (the electrode) is an input impedance in the order of $G\Omega$ [16]. Equipment with an input impedance in this order of magnitude usually cost thousands of dollars.

In articles where the authors develop their own circuits, they generally do not explain the circuits thoroughly [17,18]. In other words, the circuits and their functionality are not explained in detail, and their efficiency is not authenticated with consistent results. Besides, those circuits are usually not robust [19–21], i.e., they just amplify the signal and do not have filtering steps.

When analysing the plants' electrical response, the shape of the response depends on the type stimulus. Consequently, by analyzing the format of the excitation, along with other aspects, like propagation velocity and amplitude, we can infer which stimulus caused the response. The proposed system, i.e., the sensing step, the digital filters and the signal conditioning circuit, can be used for developing a low-cost equipment that has the purpose of monitoring and informing environmental changes, such as the ones mentioned in Section 1, in the habit of a plant.

The main contributions of this work are: to present fundamental knowledge about plants electrophysiology, focusing on the types of plants' electrical responses; the instructions to develop two types of robust signal conditioning circuits that must be selected based on the electrical signal measured. In this sense, even a user who has an intermediary familiarity with these matters, can comprehend plants electrophysiology and implement a

complete signal conditioning circuit to make the electrical response clearer before it goes to the ADC.

*1.2. Organization*

This research work is divided as follows. Section 2 presents essential information to understand the types of electrical signals that might be emitted by plants. Section 3 explains in detail the methodology employed in order to develop the entire signal conditioning circuit. Section 4 addresses the results and discussion considering Monte Carlo simulations. Finally, Section 5 presents the conclusions about the research established in the work and future works taking into account the field studied in the article.

## 2. Types of Electrical Signals

The action potential is an electrical response characterized by quickly transmitting and disseminating the disturbances along the phloem, which is one of the tissues of vascular plants, over long distances [22]. AP was the first plant's electrical response recorded and it is provoked by non-invasive excitation (e.g., electrical stimulation, thermal stress, mechanical stimulus) [11,23]. When comparing AP to Variation potential, an expressing AP attribute is that an increase in the magnitude of the excitation above a certain threshold does not modify the electrical response's shape and amplitude, as stated in [8]. One of the most important aspects of the AP is that it follows the all-or-nothing principle. To put it in other words, the tentative to cause a stimulus weaker than a certain threshold cannot trigger an action potential. Additionally, the cell membrane enters a refractory period after the period AP is triggered, in which another action potential cannot be generated or transmitted [23]. Furthermore, action potentials are able to spread through the plant structure without loss of amplitude and with constant speed, unlike VPs [1]. APs transmission speed of most plants studied previously range from 0.5 cm/s to 20 cm/s, according to [22].

Variation potential, also known as slow-wave potential, is an electrical signal generated by plants caused by damaging disturbances such as wounding, herbivore attack, and burning [23]. This signal consists of a local variation in the plasma membrane due to the transit of some other signal (chemical, hydraulic, or both combined), as stated in [24]. The xylem, which is one of the tissues of vascular plants, is the main pathway through which the VP spreads [8]. VP, unlike the AP, is defined by a decrease in the amplitude and speed of the response's propagation as it moves away from the local, which has suffered an excitation [13,24]. Besides, the plant chosen and the intensity of the disturbance influence the shape and magnitude of the VP. Variation potentials detain a great variety of changes in their shape, according to [1]. This signal can penetrate into poisoned or dead regions of the plant. Furthermore, the VP can be suppressed by a scenario of prolonged darkness and high humidity since the tension of xylem tissue becomes irrelevant, and the generation of a VP is linked to the pressure difference between the intact interior of the plant and the external environment [25]. VPs propagation speed range is from 0.1 cm/s to 1 cm/s [22].

Local electrical potential is generated at the stimuli site, which causes a sub-threshold electrical response in plants as a consequence of natural modifications in aspects connected with the external environment, such as phytohormones, fertility, and air temperature. This signal type significantly influences the plant's physiological status. The local electrical potential has a limited location, not being transmitted to other parts of the plant's body. Additionally, the intensity and duration of the excitation influence its amplitude. In addition, it can be generated using changes in the ion channel and by the transient inactivation of $H^+$-ATPase, according to [11].

System potential was first noticed by [26], being detected in leaves dozens of centimeters distant from the local that suffered the stimulus after caterpillars feeding. This signal is a self-propagate systemic signal with duration and magnitude that depends on the nature of the disturbance caused. System potential initialization is associated with the activation of $H^+$-ATPase, which induces the hyperpolarisation of the plasma membrane [11]. According to [27], this signal is strongly dependent on the conditions and treatments of the

experiments. Besides, different from AP, SP does not follow the all-or-none rule. Weak stimuli that are not enough to initiate APs, since they do not reach the critical intensity, can trigger system potentials. Additionally, SP is triggered by a hyperpolarization of the plasma membrane. This is unlike AP and VP, which begin with a depolarization. System potential has a propagation speed that ranges from 5 cm/min to 10 cm/min [11].

### 3. Proposed Methodology

Figure 3 illustrates the steps of the proposed methodology. This diagram shows two signal conditioning circuits options, each for a different frequency range. Since plants' electrical responses have frequency components that range from very low frequencies [28,29] to several hundreds of Hertz, according to [30], it is necessary to design the conditioning circuit taking into account the measured signal range. Plants' signal frequency depends on their species, growth stage, measured tissue, and the excitation source.

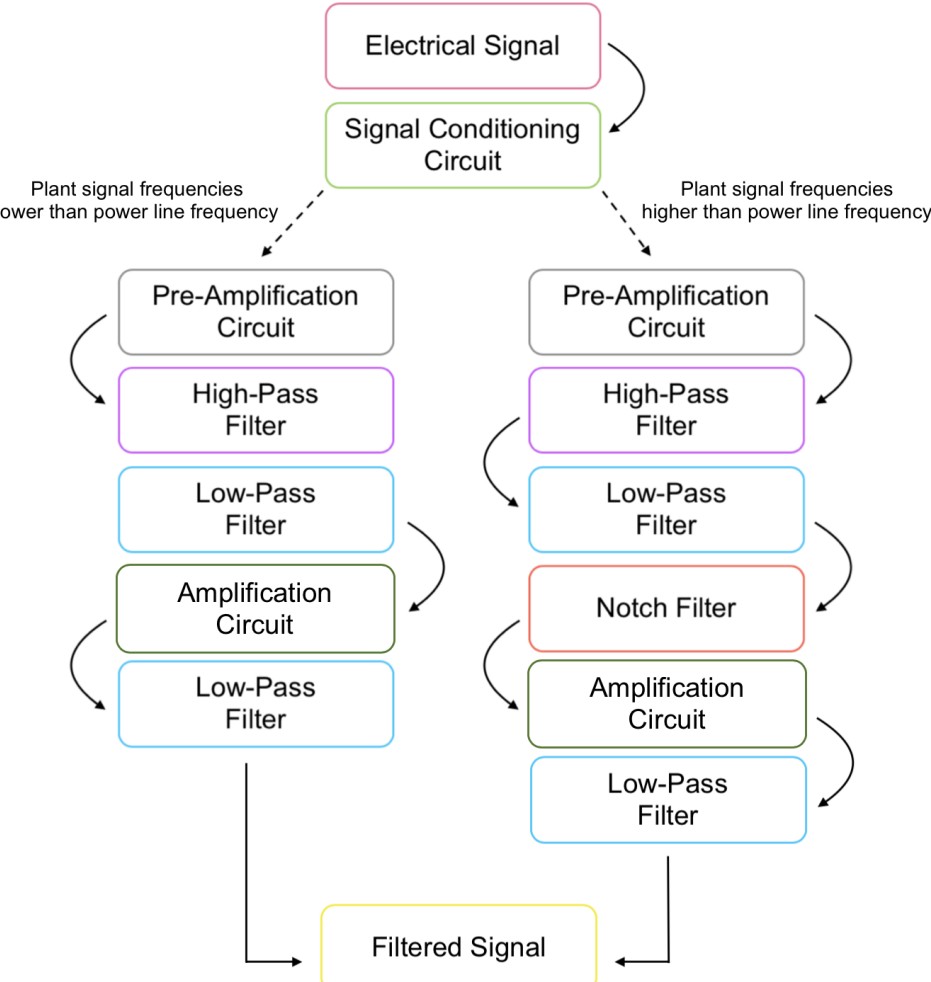

**Figure 3.** Flowchart of the proposed methodology.

Electrical signals generated by plants have low amplitude, in the order of tens of μV to tens of V [28]. So, a signal conditioning circuit is crucial to improve the SNR of the electrical response. SNR compares the level of a desired signal with respect to the level of background noise.

Moreover, the electrical response amplitude must fit within the ADC dynamic range. An ADC's dynamic range is the range of signal amplitudes which the analog-to-digital converter is able to resolve. To use the full resolution provided by the ADC, the input signal range must be the same as the ADC operation range.

The electrical signal conditioning circuit structures presented in this work are shown in Figures 4 and 5.

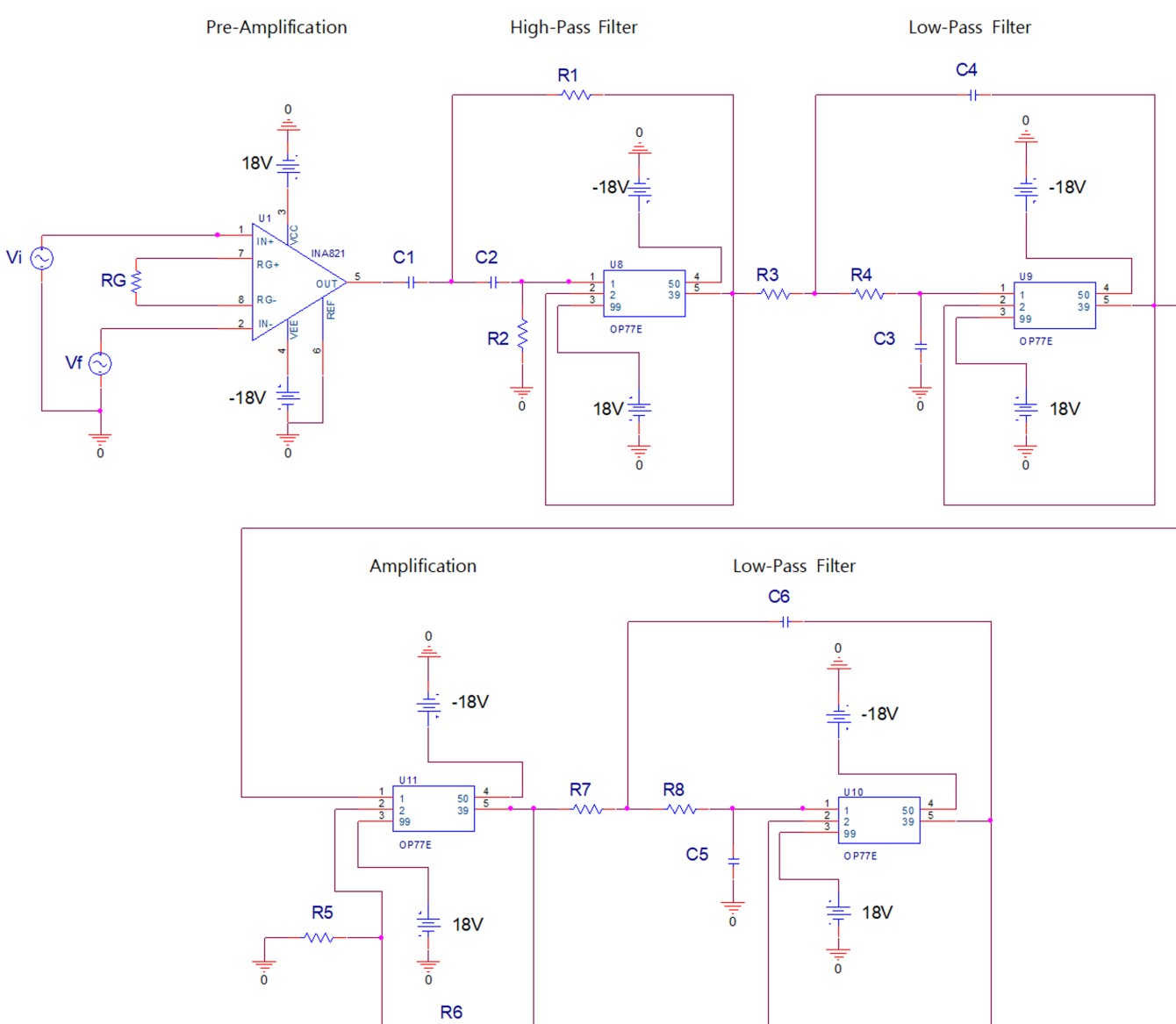

**Figure 4.** Schematic diagram of the first developed signal conditioning circuit. In the pre-amplification stage, an INA821 instrumentation amplifier is applied. In the high-pass and low-pass filters steps, and in the amplification stage, an OP77 op-amp is employed as well.

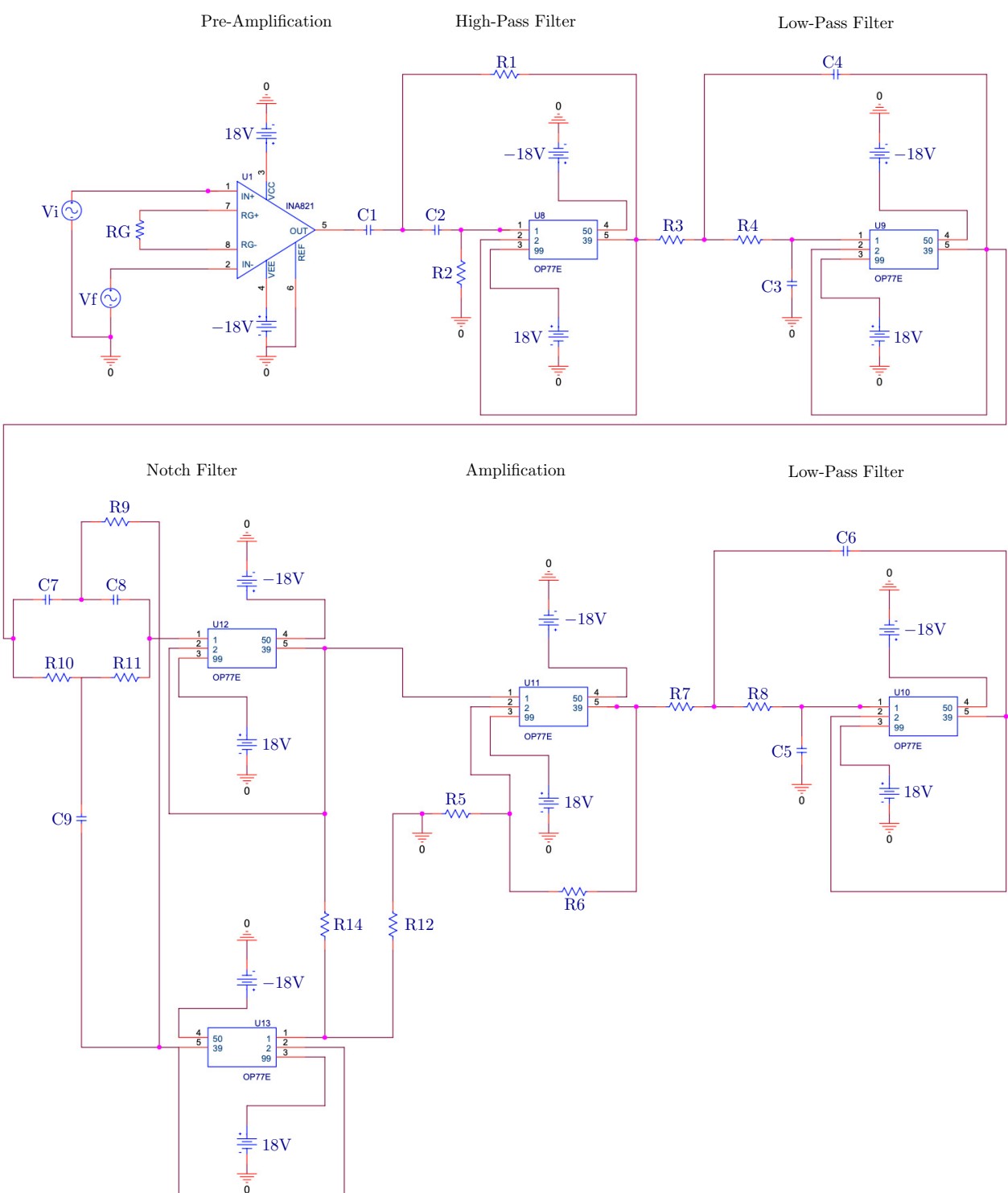

**Figure 5.** Schematic diagram of the second developed signal conditioning circuit. In the pre-amplification stage, an INA821 instrumentation amplifier is employed. In the high-pass, low-pass, and notch filters steps, and in the amplification stage, an OP77 op-amp is employed as well.

### 3.1. Pre-Amplification Circuit

The pre-amplification stage is the most crucial of the entire signal conditioning circuit because, if it is adequately built with a differential amplifier, an appreciable part of the common mode noise that interferes with the plant's electrical response can be minimized. The input impedance of the pre-amplification circuit must be in the order of G$\Omega$. The reason for this input impedance order is that the impedance value of $Ag/AgCl$ electrodes, which

are the most commonly used electrodes in this application, is in the order of a few kΩ, and the source impedance (plant) often has a value in the order of hundreds or thousands of kΩ [31]. As a consequence, the value of the electrical response that appears at the input of the pre-amplification step is approximately equal to the plant signal if the circuit input impedance is as large as possible.

The input offset voltage temperature coefficient of the op-amp employed in the pre-amplification circuit has to be less than 10 µV/°C, as in [17]. Usually, the gain value applied to the signal in this stage range from 10 to 50.

Additionally, as stated in [18], the common-mode rejection ratio has to be at least 100 dB, so the power line frequency interference, which is present on the non-inverting and inverting op-amp inputs, can be attenuated adequately [32]. This parameter indicates how much an undesired common-mode signal influences the measurements, which is a crucial criterion.

In the pre-amplification stage, it is recommended to employ the classic instrumentation amplifier structure using three op-amps, as used in [32]. In this classic architecture there are two amplifiers in the voltage follower configuration with a third op-amp, as in [17], or an instrumentation amplifier integrated circuit. The differential amplifier configuration employing only one op-amp cannot be used at this step for the sake of does not offer the necessary input impedance. Some instrumentation amplifier integrated circuits that can be applied in the first step are AD8221, INA821 and INA128, as stated in [12]. INA821 was chosen to be used in the simulation, and the expression related to the gain of this stage can be seen in Equation (1).

$$G = 1 + \frac{49.4 \text{ k}\Omega}{R_G} \tag{1}$$

The output of the pre-amplification circuit is given by Equation (2).

$$V_{OUT} = G(V_{IN+} - V_{IN-}) + V_{REF} \tag{2}$$

### 3.2. Low-Pass and High-Pass Filters

Sallen-Key configuration, which is non-inverting, is applied to the second and third stages. Sallen-Key filter topology was selected because it is a low-complexity second-order filter, and it is the least dependent on the frequency response of the chosen op-amp, according to [33]. The second stage consists of a high-pass filter, and the third one is a low-pass filter, both of which have unit gain. OP07 and OP77 are op-amps that can be employed in these steps. Moreover, it is feasible to use some possible filter approximations, like Butterworth, Chebyshev, and Bessel, depending on the adjustment of the quality factor $Q$. It is important to cite that these approximations dictate the format of the frequency response.

A bandpass filter was made by cascading a high-pass filter with a low-pass filter opposite to applying only one op-amp. The advantage of customizing the filter to have an asymmetrical response is the motivation for this procedure. A Sallen-Key bandpass filter using only one op-amp has got cut-off frequencies symmetrically apart from the center frequency $f_0$. Note that OP77 was selected to be applied in the simulation. The Sallen-Key equations for the high-pass filter are given by Equations (3)–(5), which represent the transfer function respectively, $f_c$, and $Q$ [12].

$$\frac{V_{39}}{V_1} = \frac{s^2(R_1 R_2 C_1 C_2)}{s^2(R_1 R_2 C_1 C_2) + s R_1(C_1 + C_2) + 1} \tag{3}$$

$$f_c = \frac{1}{2\pi\sqrt{R_1 R_2 C_1 C_2}} \tag{4}$$

$$Q = \frac{\sqrt{R_1 R_2 C_1 C_2}}{R_1(C_1 + C_2)} = \frac{1}{2\pi f_c R_1(C_1 + C_2)} \tag{5}$$

Sallen-Key equations for low-pass filter are given by Equations (6)–(8), which represent respectively the transfer function, $f_c$, and $Q$ [12].

$$\frac{V_{39}}{V_1} = \frac{1}{s^2(R_3 R_4 C_4 C_3) + sC_3(R_3 + R_4) + 1} \tag{6}$$

$$f_c = \frac{1}{2\pi\sqrt{R_3 R_4 C_4 C_3}} \tag{7}$$

$$Q = \frac{\sqrt{R_3 R_4 C_4 C_3}}{C_3(R_3 + R_4)} = \frac{1}{2\pi f_c C_3(R_3 + R_4)} \tag{8}$$

### 3.3. Notch Filter

The notch filter step is only used in the signal conditioning circuit shown in Figure 5, which is employed for plants' signals with frequency components higher than the power line frequency. Notch filters are part of a special class of band-stop filters capable of rejecting a very narrow range of frequencies. It acts almost exclusively on the selected frequency, in this case, the power line frequency.

The one applied in this project was Twin-T Notch Active Filter, and it was chosen instead of this filter's passive implementation because the latter has a significant shortcoming of a $Q$ fixed at 0.25 [33]. The active configuration holds a variable $Q$, allowing the user to set its value in a way that can achieve the best compromise between rejection at the notch frequency $f_n$ and bandwidth $BW$ since these two variables are related by Equation (9).

$$Q = \frac{f_n}{BW} \tag{9}$$

The amount of the signal feedback determines the value of $Q$ of the circuit, which, in turn, defines the notch depth. This parameter is set by $R_{14}/R_{12}$ ratio. The design equations for the Twin-T notch filter are shown in Equations (10) and (11) [33]. $V_{39'}$ is the input and refers to the output of the low-pass filter.

$$\frac{V_{39}}{V_{39'}} = \frac{s^2 + \omega_o^2}{s^2 + \frac{s\omega_o}{Q} + \omega_o^2} = \frac{s^2 + \left(\frac{1}{RC}\right)^2}{s^2 + s\left(\frac{1}{RC}\right)\left(\frac{4}{1 + \frac{R_{12}}{R_{14}}}\right) + \left(\frac{1}{RC}\right)^2} \tag{10}$$

$$f_n = \frac{1}{2\pi RC} \tag{11}$$

OP77 was chosen to be employed in the simulation but OP07 and TLV2252ID [34] are op-amps that can be applied in this step.

### 3.4. Amplification Circuit

The last but one stage of the signal conditioning circuit includes a non-inverting configuration in that the gain is determined from the selected resistor values. Normally, the gain value applied to the signal in this stage range from 10 to 1000. It is needful to point out that the higher the gain value, the narrower the bandwidth op-amp will work without the signal being attenuated. Therefore, it is important to guarantee the bandwidth the op-amp is working with a certain gain covers all frequencies of the plant's signal selected to perform measurements without suffering attenuation. OP07 and OP77 are op-amps that can be used in this amplification stage [12]. The mathematical statements related to this stage can be seen in Equations (12)–(16).

$$V_1 = V_2 = V_{in} \tag{12}$$

Resulting in:

$$V_{39} = V_{out} \tag{13}$$

$$\frac{V_{in} - 0}{R_5} + \frac{V_{in} - V_{out}}{R_6} = 0 \tag{14}$$

$$V_{in}R_6 + V_{in}R_5 - V_{out}R_5 = 0 \tag{15}$$

The gain is defined by:

$$A_v = 1 + \frac{R_6}{R_5} \tag{16}$$

### 3.5. Anti-Aliasing Filter

In conclusion, the last step of the circuit is the anti-aliasing filter, which is a low-pass filter with the cut-off frequency set to the Nyquist frequency. Sallen-Key low-pass topology is used in this step too. Besides, OP77 and OP07 are op-amps that can be employed in this stage.

At the end of the whole process, the electrical signal shows up clearer at the signal conditioning circuit output, stronger and with undesired frequencies attenuated, ready for the ADC and digital filtering step.

## 4. Results and Discussion

The signal conditioning circuits, shown in Figures 4 and 5, were simulated in OrCAD Capture 16.6 software to testify the functionality for which they were proposed. The values chosen for the components are shown in Tables 1 and 2.

**Table 1.** Components values of the first signal conditioning circuit.

| Components | Values | |
|---|---|---|
| Resistors | $R_G = 2.4 \text{ k}\Omega$ $R_1 = 39 \text{ k}\Omega$ $R_2 = 82 \text{ k}\Omega$ $R_3 = 10 \text{ k}\Omega$ $R_4 = 10 \text{ k}\Omega$ | $R_5 = 1 \text{ k}\Omega$ $R_6 = 100 \text{ k}\Omega$ $R_7 = 5 \text{ k}\Omega$ $R_8 = 5 \text{ k}\Omega$ |
| Capacitors | $C_1 = 5.6 \text{ μF}$ $C_2 = 5.6 \text{ μF}$ $C_3 = 0.27 \text{ μF}$ | $C_4 = 560 \text{ nF}$ $C_5 = 0.28 \text{ μF}$ $C_6 = 0.56 \text{ μF}$ |

**Table 2.** Components values of the second signal conditioning circuit.

| Components | Values | |
|---|---|---|
| Resistors | $R_G = 2.4 \text{ k}\Omega$ $R_1 = 39 \text{ k}\Omega$ $R_2 = 82 \text{ k}\Omega$ $R_3 = 3 \text{ k}\Omega$ $R_4 = 3 \text{ k}\Omega$ $R_5 = 1 \text{ k}\Omega$ $R_6 = 100 \text{ k}\Omega$ | $R_7 = 3 \text{ k}\Omega$ $R_8 = 3 \text{ k}\Omega$ $R_9 = 13 \text{ k}\Omega$ $R_{10} = 27 \text{ k}\Omega$ $R_{11} = 27 \text{ k}\Omega$ $R_{12} = 80 \text{ k}\Omega$ $R_{14} = 20 \text{ k}\Omega$ |
| Capacitors | $C_1 = 5.6 \text{ μF}$ $C_2 = 5.6 \text{ μF}$ $C_3 = 0.37 \text{ μF}$ $C_4 = 0.75 \text{ μF}$ $C_5 = 0.18 \text{ μF}$ | $C_6 = 0.37 \text{ nF}$ $C_7 = 0.1 \text{ μF}$ $C_8 = 0.1 \text{ μF}$ $C_9 = 0.2 \text{ μF}$ |

Furthermore, Monte Carlo simulations were carried out employing the same software with the intention of checking the behavior of the circuits, taking into account possible variations in the nominal components value.

### 4.1. Filters Frequency Response

In the second stages of Figures 4 and 5, values of capacitors and resistors of the high-pass filters were selected so that they could have $Q = 0.707$ and $f_c = 0.5$ Hz. In the third stages of Figures 4 and 5, which are low-pass filters, values of the capacitors and resistors were chosen so they could have $Q = 0.707/f_c = 40$ Hz and $Q = 0.707/f_c = 100$ Hz, respectively. In the last stages, which are anti-aliasing filters, values of resistors and capacitors were selected so that they could have $f_c = 100$ Hz$/Q = 0.707$ and $f_c = 200$ Hz$/Q = 0.707$, respectively. Taking into account the notch filter, the $f_n$ chosen was 60 Hz, because this is the power line frequency employed in Brazil. Besides, the $Q$ value is 2.5.

The cut-off frequencies of Figure 4 were chosen to take into account a plant electrical signal with frequency components between 5 Hz and 25 Hz [18,35,36]. Additionally, the cut-off frequencies of Figure 5 were set considering a plant electrical signal with frequency components between 5 Hz and 85 Hz [30]. It is important to highlight that it is not suggested to choose the $f_c$ exactly equal to the minimum and maximum frequencies components of the signal to be measured. Note that the user commonly does not know the minimum/maximum frequency components of a determined electrical response of a specific plant. If the low-pass filter $f_c$ set with the slack is higher than the power line frequency, it will be necessary to apply the circuit of Figure 5. Even if the signal to be measured is supposed to have frequency components lower than the power line frequency. Figures 6–8 show the magnitude responses for the high-pass, notch and low-pass ($f_c = 100$ Hz) filters, respectively.

Taking into account Figure 6 high-pass filter, it is needful to cite that the cut-off frequency found in the simulation was $f_c = 0.517$ Hz. The notch frequency achieved for the notch filter shown in Figure 7 was $f_n = 63.096$ Hz. Considering Figure 8 low-pass filter, the cut-off frequency obtained in the simulation was $f_c = 101.141$ Hz. For the low-pass filters with cut-off frequencies equal to 200 Hz, 80 Hz, and 40 Hz, the simulation results provided $f_c = 203.667$ Hz, 80.157 Hz, and 41.517 Hz, respectively.

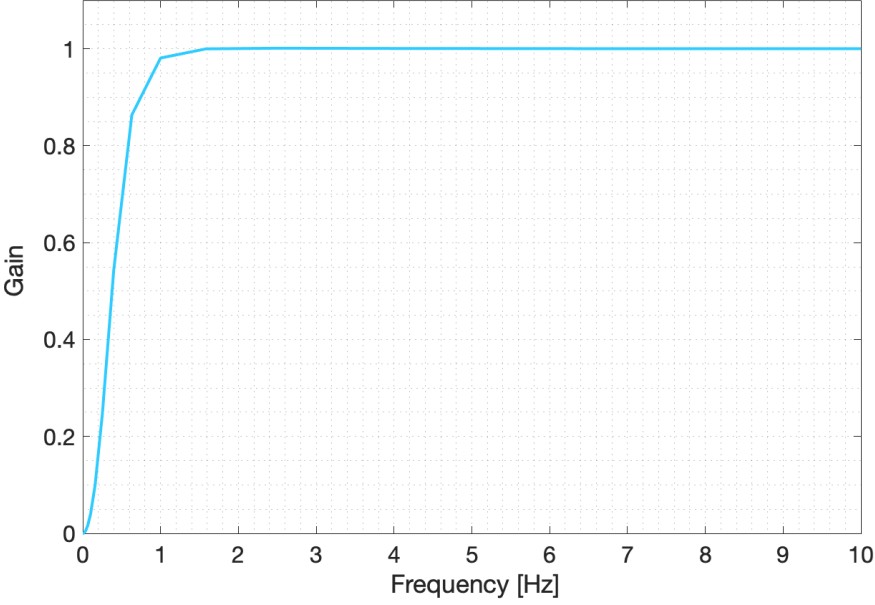

**Figure 6.** Magnitude response for the high-pass filter. The plot shows the filter's gain for different sinusoidal inputs frequencies. The filter is configured to reject frequencies lower than $f_c = 0.5$ Hz.

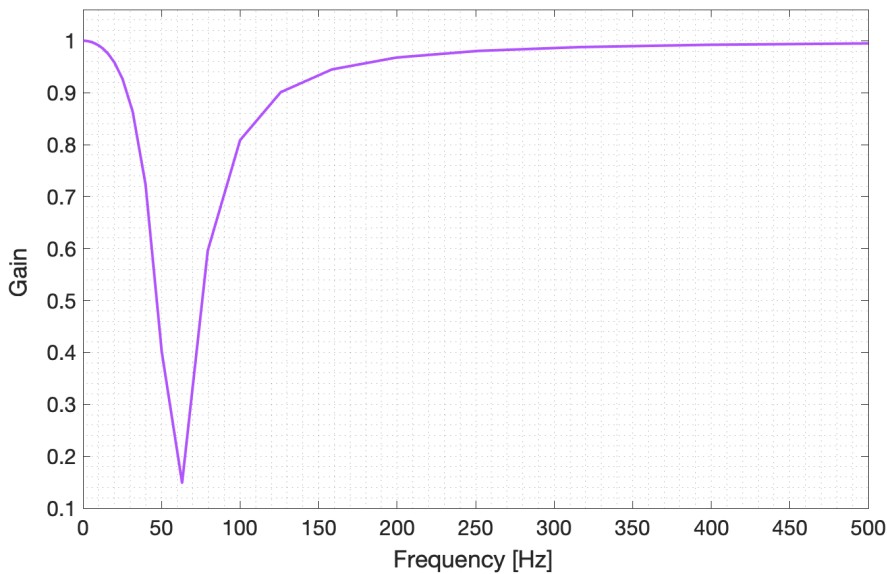

**Figure 7.** Magnitude response for the notch filter. The plot shows the filter's gain for different sinusoidal inputs frequencies. The filter is configured to reject the power line frequency of $f_n$ = 60 Hz.

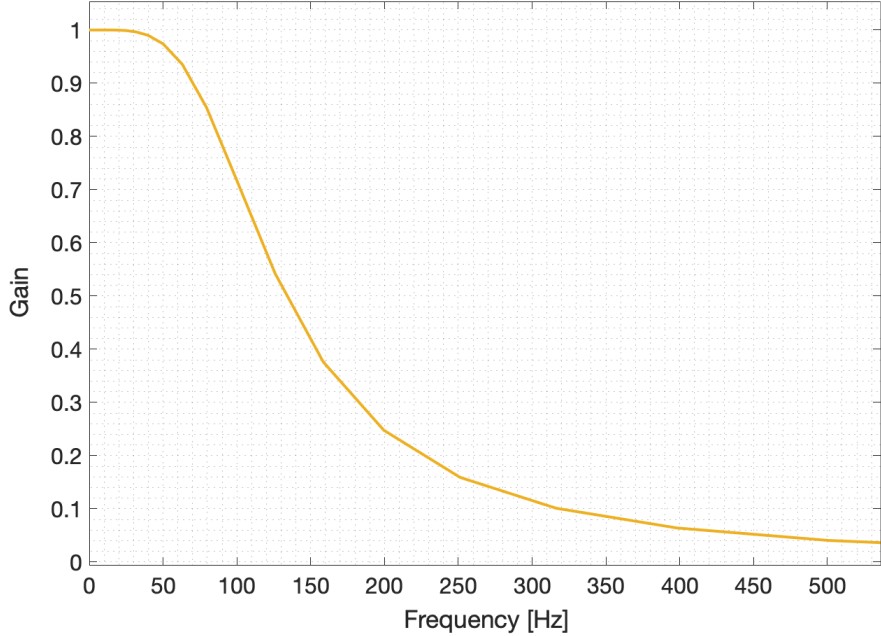

**Figure 8.** Magnitude response for the low-pass filter. The plot shows the filter's gain for different sinusoidal inputs frequencies. The filter is configured to reject frequencies higher than $f_c$ = 100 Hz.

### 4.2. Signal Conditioning Circuit Simulation

In the pre-amplification stage, a gain of 21.58× (26.68 dB) was set, and $V_{REF} = 0$. In the stage in which the electrical signal is amplified, values of the resistors were selected employing Equation 16 so that the gain setting could be 101× (40.09 dB). Figures 9 and 10 show the gain of each stage of circuits 1 and 2, respectively, when they are submitted to sinusoidal inputs of varying frequencies. These figures show the filters cascade response, meaning that each stage refers to the output of that stage and all previous stages combined. At the first stage, pre-amplification, it is possible to see that the gain value of 26.69 dB given to the differential signal at the input ($V_{IN+}$-$V_{IN-}$) is constant for the range of tested frequencies and is very close to what was expected. Then, at the high-pass filter stage, which is the cascade of the pre-amplification circuit and a high-pass filter, both Figures 9 and 10 show that the gain stays the same, but the high-pass filter introduces a cut-off frequency

at 0.49 Hz. Following the cascade, at the low-pass filter stage, the gain and lower cut-off frequencies remain unaltered. Figures 9 and 10 show that the low-pass filter introduces high cut-off frequency at 41.56 Hz and 101.1 Hz, respectively. At the end of the low-pass filter stage, we can see that, due to the filter cascade, both circuits work as band-pass filters with a gain defined by the pre-amplification step and cut-off frequencies defined by the low-pass and high-pass filters stages.

As can be seen in Figure 9, for circuit 1, the step after the low-pass filter stage is the amplification step, which adds a gain of 40.09 dB. Due to the cascade of filters, the total gain at the amplification stage is 66.78 dB (the sum of the pre-amplification and amplification steps). Moreover, for circuit 1 (Figure 9), the last stage, anti-aliasing filter, is a low-pass filter which has a cut-off frequency (theoretically 100 Hz) higher than the previous low-pass filter in the cascade (41.56 Hz). Consequently, the result of the cascade has a high cut-off frequency, smaller than both low-pass filters, at 40.3 Hz.

For circuit 2, as can be seen in Figure 10, the stage that follows the low-pass filter step is the notch filter. At this stage, the notch filter introduces a rejection peak in which the minimum is at 59.5 Hz. Similarly to circuit 1, at the amplification stage of circuit 2, the total gain is 66.78 dB due to the cascade of amplifications. Finally, in the last stage, the anti-aliasing filter has a similar frequency to the low-pass filter introduced earlier in the cascade (100 Hz versus 101.1 Hz). Therefore, the result of the cascade has a similar shape before and after the anti-aliasing filter, but the gain at frequencies higher than 100 Hz decreases faster with respect to the increase in frequency.

In summary, Figures 9 and 10 show that when input frequencies are within the passband of the high-pass and low-pass filters, the electrical signal is amplified throughout the stages, and is attenuated otherwise. Taking into account the notch filter of Figure 10, frequencies around the 60 Hz notch frequency are also rejected. As stated in [15] for plants' electrical signals, voltage values employed in the circuits are in the range from tens of μV to tens of V.

In Section 4.1, the filters' frequency response was presented when each filter was simulated individually. For both signal conditioning circuit designs (with or without the notch filter), the filters are connected in cascade, as shown in Figures 4 and 5. When two filters are connected, the first output serves as the second's input. Because of this, the input of the second filter is already modified by the first filter. As a result, when two low-pass filters are attached, as in circuit 1, the cut-off frequency of the entire cascade is not the same as one of the filters' cut-off frequencies.

The tolerance of the components may influence the gain given in the first and last stages. Moreover, this parameter can influence $Q$ and $f_c$ of the filters because they are dependent on component values.

Due to their maximally flat magnitude response in the passband, Butterworth filters were chosen to be applied. However, this filter type introduces a customarily undesired phase shift into the filtered data, as shown in Figures 11 and 12. The delay length elevates with increasing filter order and decreasing $f_c$.

In Figure 11, the amplitude of the first, second, and third stages are similar to each other. Moreover, their amplitude is much lower than in the following stages since the gain given to the electrical signal in the fourth stage is 40 dB. As a result, the first, second, and third stages graphs are not clearly shown in the figure. The same situation occurs in Figure 12 because the first four stages have lower amplitude than the fifth and sixth stages.

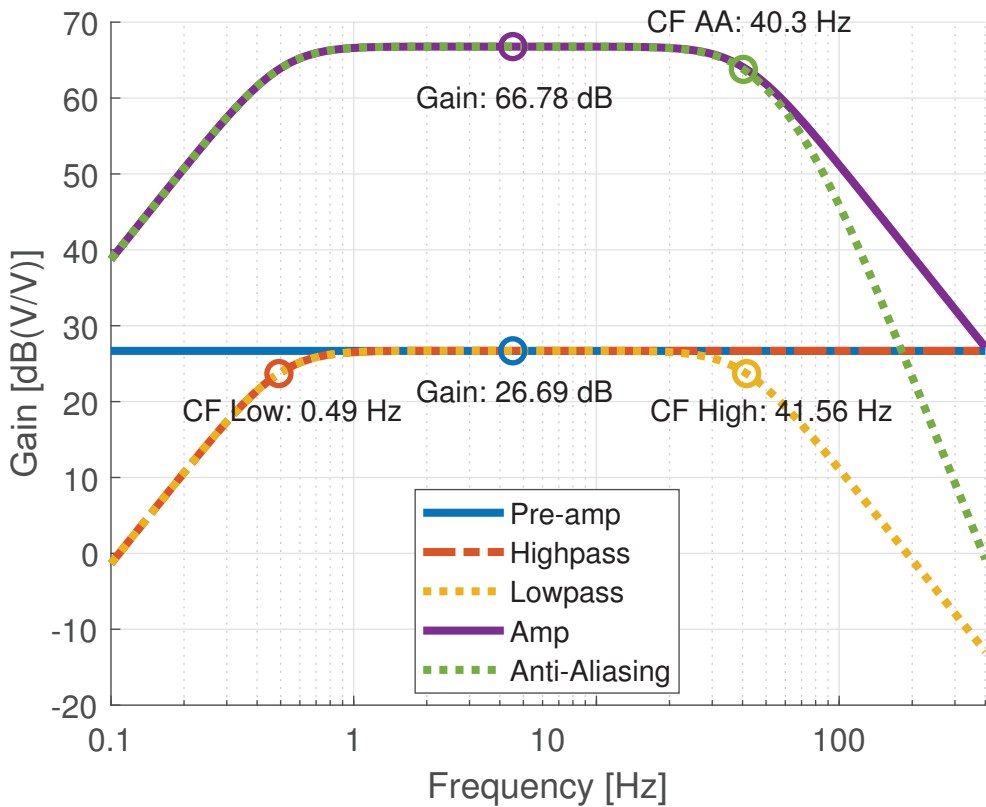

**Figure 9.** Magnitude response of all stages of circuit 1. "CF" stands for cut-off frequency and "AA" stands for anti-aliasing.

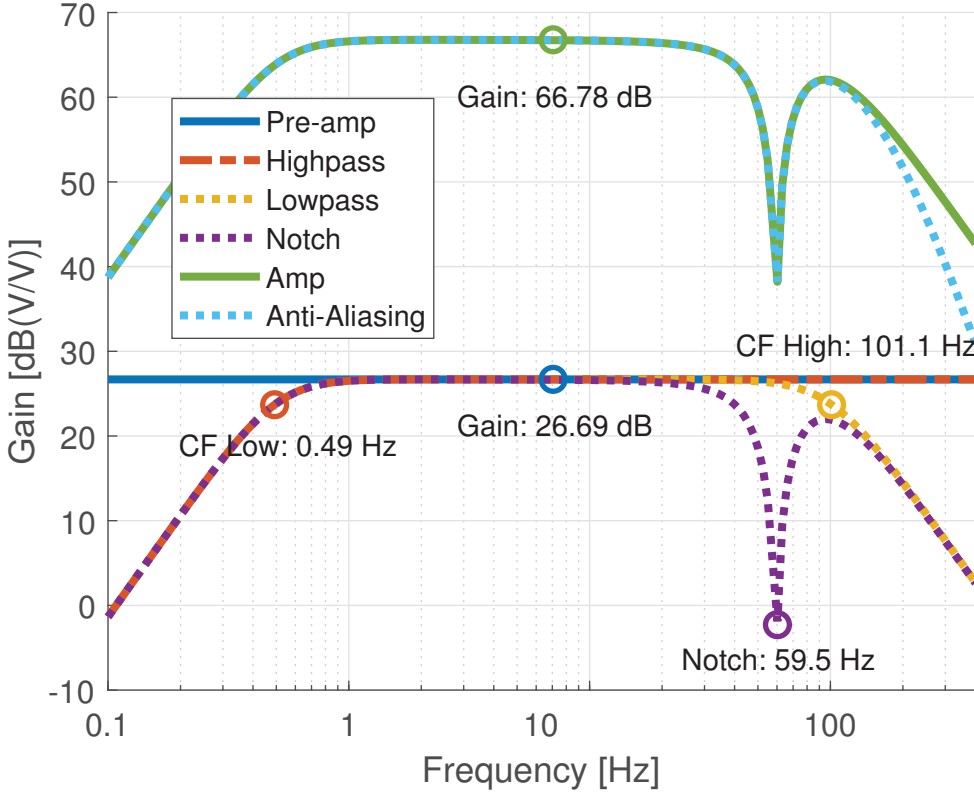

**Figure 10.** Magnitude response of all stages of circuit 2. "CF" stands for cut-off frequency and "Notch" refers to notch frequency.

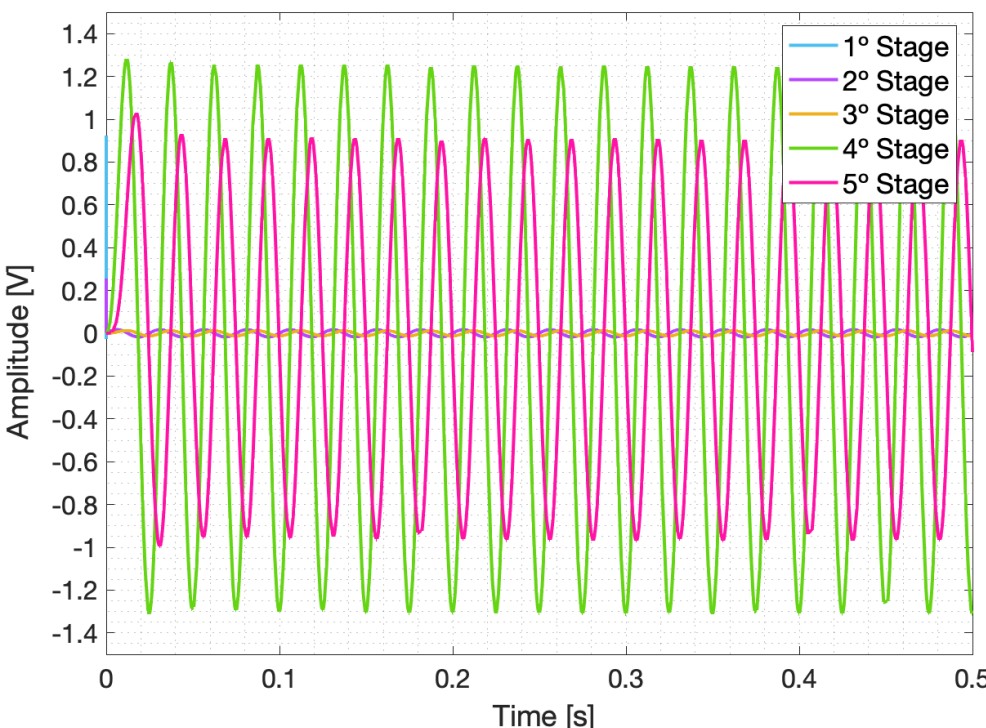

**Figure 11.** Graphs of each conditioning circuit 1 stage when *f* = 40 Hz.

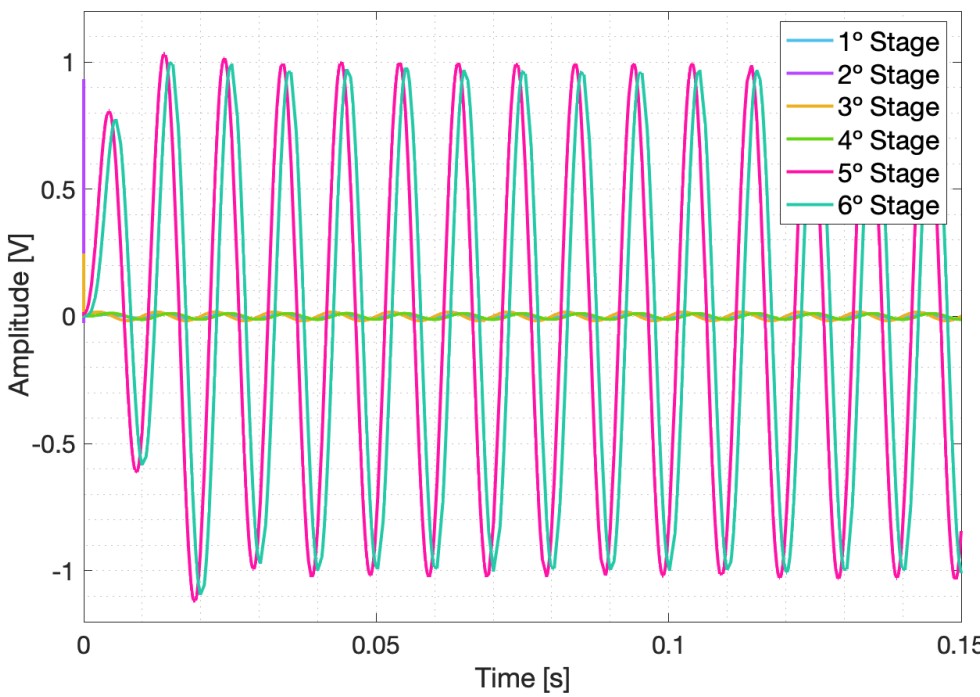

**Figure 12.** Graphs of each conditioning circuit 2 stage when *f* = 100 Hz.

*4.3. Components Variation Simulation*

Electronic components have a nominal, as labeled by the manufacturer, and a real value. There are imperfections in manufacturing these components; therefore, their real value is not always the same as the nominal value. Simulations can be performed to verify the behavior of a circuit, taking into consideration possible variations in the nominal value of components.

To further characterize circuits 1 and 2, and validate their performance for real-world cases, Monte Carlo simulations are performed. These simulations use a given statistical distribution to slightly alter the value of each component within the specified tolerance range. Each Monte Carlo sample refers to a possible set of random component values. To produce statistically relevant results, usually, hundreds of samples are simulated.

In this work, the model used for the Monte Carlo simulations is the default model provided by OrCAD, and is defined as follows. Resistors and capacitors have their values independently randomized following a Gaussian distribution for each circuit. The distribution is adjusted so that the resulting values (after being randomized) fall within the components' 1% tolerance. A different Gaussian distribution is generated for each component, which has a mean equal to the respective nominal component value and with three standard deviations being the nominal value after 1% variation. A hundred Monte Carlo samples are simulated for each circuit.

Figure 13 shows the magnitude response for the last stage of circuit 1. Each Monte Carlo sample represents a different set of component values and, thus, generates a different magnitude response. For circuit 1, the required (without component value variation) output behavior is a band-pass filter. As can be seen in Figure 13, all Monte Carlo variations have the demanded output behavior.

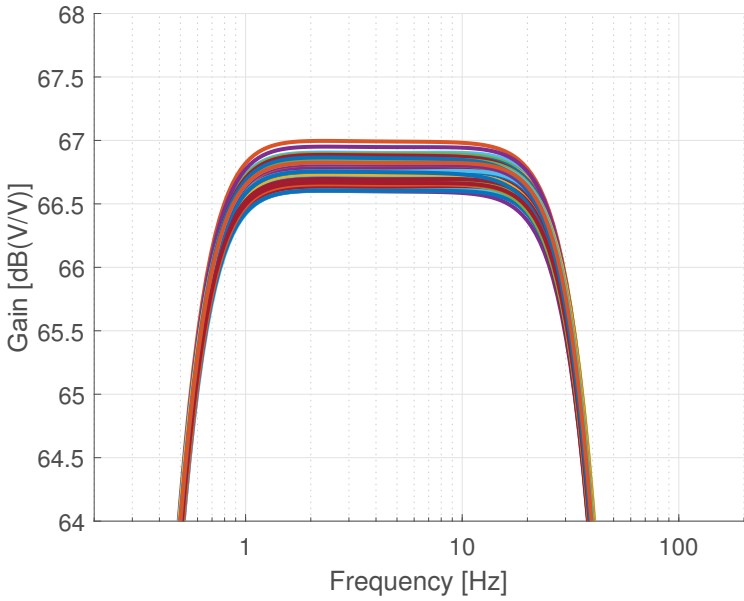

**Figure 13.** Magnitude response of the Monte Carlo simulation samples of the last stage of circuit 1. Each line plot represents a different Monte Carlo sample.

To closely examine Monte Carlo samples, for each sample of circuit 1, the lower and higher cut-off frequencies are computed. When observing all Monte Carlo samples of circuit 1's output, the lower cut-off frequency holds an average value of 0.49 Hz and variance of $1.1 \times 10^{-5}$, and the higher cut-off frequency has an average value of 40.39 Hz and variance of $1.8 \times 10^{-1}$.

Figure 14 presents the distribution of both cut-off frequencies. The computed frequencies are normalized regarding their respective mean to show the relative variation between samples better. As can be seen in Figure 14, for the simulations performed, the lower and higher cut-off frequencies have got at most 2% and 4% variation from the mean, respectively. However, most Monte Carlo samples resulted in frequencies within less than 1% variation.

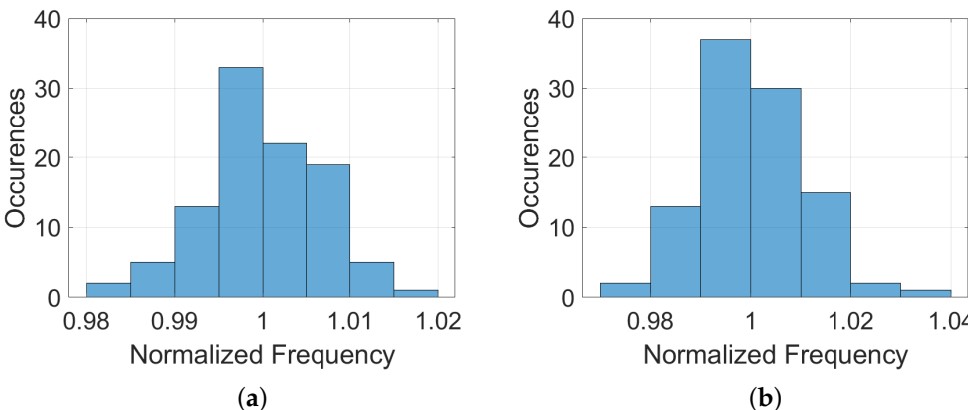

**Figure 14.** Histogram of the (**a**) lower and (**b**) higher cut-off frequency of the last stage of circuit 1. Both frequencies are normalized with respect to the mean of all 100 Monte Carlo samples of each respective frequency.

Figure 15 shows the magnitude response for the last stage of circuit 2. The Monte Carlo samples appear more similar to each other when compared to circuit 1, because of the plot scale. For circuit 2, the desired output behavior is also a band-pass filter, but with a notch filter with $f_n$ = 60 Hz. As shown in Figure 15, all Monte Carlo variations have the desired output behavior.

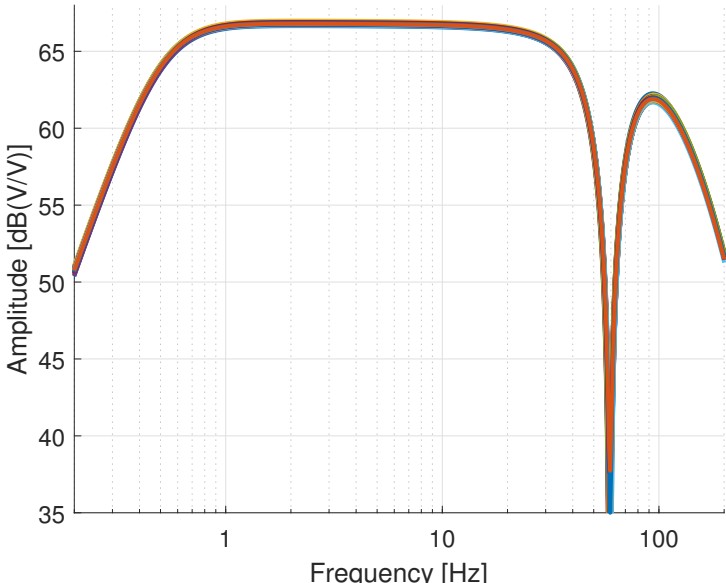

**Figure 15.** Magnitude response of the Monte Carlo simulation samples of the last stage of circuit 2. Each line plot represents a different Monte Carlo sample.

When observing all Monte Carlo Samples of circuit 2, the lower cut-off frequency has a mean value of 0.49 Hz and variance of $1.2 \times 10^{-5}$ and the center frequency of the notch stopband has a mean value of 59.46 Hz and variance of $1.2 \times 10^{-1}$.

Figure 16 shows the distribution of the notch filter's lower cut-off frequency and center frequency for circuit 2's output. As can be seen in Figure 16, for the simulations performed, the lower cut-off and notch frequencies have at most 2% and 1.5% variation from the mean, respectively. However, most Monte Carlo samples resulted in frequencies within less than 1% variation.

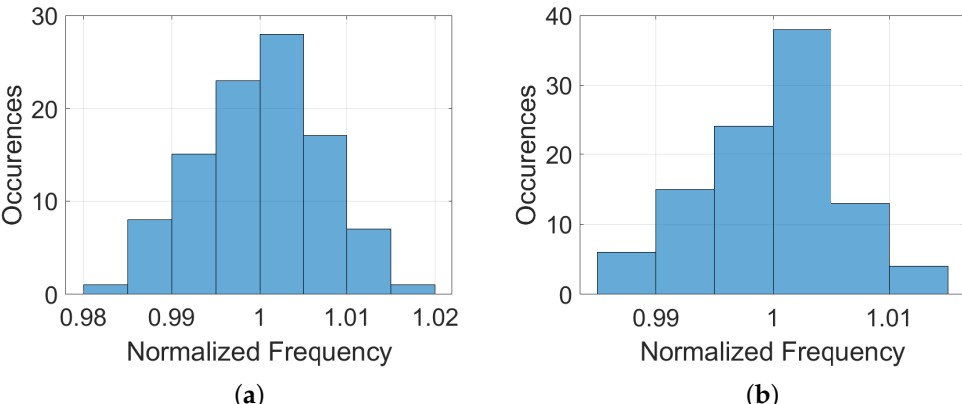

**Figure 16.** Histogram of the (**a**) lower cut-off frequency and (**b**) center frequency of the notch band-stop filter of the last stage of circuit 2. Both frequencies are normalized with respect to the mean of all 100 Monte Carlo samples of each respective frequency.

## 5. Conclusions and Future Work

This work has presented valuable signal conditioning circuits that operate efficiently. Computer simulations allowed the authors to validate the circuits' behavior via software without carrying out bench tests. The results obtained with this project are similar to the ones expected by the theory. The methodology presented can be followed and adjusted according to the type of plant and its electrical signals. In addition, through Monte Carlo simulations, OrCAD Capture software is able to generate hundreds of possible variations in the circuit's parameters. With this approach, results that more closely resemble real-world performance were also showed. This information makes it possible to determine which circuits are suitable for the required application.

Therefore, this work opens the possibility of several improvements in terms of implementation. In future works, the authors intend to include experiments of these signal conditioning circuits using different species of plants, employing the project developed in this work.

**Author Contributions:** Conceptualization, M.C. and V.O.; methodology, M.C., V.O. and M.P.; software, M.C. and V.O.; validation, M.C. and V.O.; formal analysis, M.C. and V.O.; investigation, M.C. and V.O.; resources, F.O., M.C., V.O. and M.T.; data curation, M.C. and V.O.; writing—original draft preparation, M.C., V.O. and M.P.; writing—review and editing, M.C., V.O., F.O. and M.P.; visualization, M.P. and M.T.; supervision, M.P., F.O. and M.T.; project administration, F.O. and M.P.; funding acquisition, M.T. All authors have read and agreed to the published version of the manuscript.

**Funding:** This research received no external funding.

**Data Availability Statement:** Not applicable.

**Acknowledgments:** The authors would like to thank UFRJ, CEFET/RJ, UTFPR and the brazilian research agencies CAPES, CNPq, and FAPERJ.

**Conflicts of Interest:** The authors declare no conflict of interest.

## Abbreviations

The following abbreviations are used in this manuscript:

| | |
|---|---|
| ADC | Analog-to-Digital Converter |
| APs | Action Potentials |
| LEPs | Local Electrical Potentials |
| SNR | Signal-to-Noise Ratio |
| SPs | Systems Potentials |
| VPs | Variation Potentials |

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
