# Peer review of "Simulation Analysis of Signal Conditioning Circuits for Plants’ Electrical Signals"

_technologies, doi:10.3390/technologies10060121_

Round 1

Reviewer 1 Report

The authors addressed an interesting topic appropriate for the journal.

By my side I recommend the following improvements:

- to introduce the concept of plant technology in more detail (with illustrative images, extracted also from other papers);

- to better discuss figure 9; does the peak represent an antiresonance? It looks like a pole of the transfer function;

- to introduce the parameters used in the Monte Carlo simulations and of the developed model;

- is there an experimental validation of the proposed model?

Author Response

Dear Reviewers,

First, the authors would like to thank the opportunity to further improve this manuscript. We appreciate your attention and careful review. We have implemented all the requested changes. To facilitate the review, we use this color scheme:

• Blue: Author’s answer
• Red: Modifications in the manuscript.

All answers are in the attached PDF.

Best regards, Authors.

Author Response

(The authors gave the same response as above.)

Round 2

Reviewer 1 Report

Dear authors,

for me the paper is well done.

My best regards.